# An Integrated Multi-Attribute Model for Evaluation of Sustainable Mobile Phone

**Morteza Yazdani** [1,*] **, Prasenjit Chatterjee** [2] **, Maria Jose Montero-Simo** [1] **and Rafael A. Araque-Padilla** [1]

1   Department of Management, Universidad Loyola Andalucía, 41014 Sevilla, Spain
2   Department of Mechanical Engineering, MCKV Institute of Engineering, Howrah 711204, India
*   Correspondence: myazdani@uloyola.es; Tel.: +34-955-641-600

**Abstract:** Consumer preferences in sustaining and designing a product are a vital driver in a company's long-term strategy. In a supply chain management (SCM), realizing, configuring and analyzing consumer point of view and making sure the product is highly fitted to the consumer dimensions are essential responsibilities. For this purpose, a sustainable supply chain (SSC) can define a platform in order to reach consumer satisfaction. This paper examines the utility and factors related to the use of a phone in the market incorporating sustainable attributes. We firstly identify main factors and indicators that influence the selection of a sustainable phone. Thereafter, we propose decision analysis tools as decision-making trial and evaluation laboratory (DEMATEL) and analytical hierarchy process (AHP) for the realization of the cause, effect, and interrelation of the indicators. The comparisons between them report a high similarity while best and worst indicators are in the same positions. Best worst method (BWM) is then formulated in order to achieve optimal ranking and to express the importance. Counting on this information is of special relevance in marketing decision-making, where companies must look for competitive advantages prioritizing its product attributes, attending both to resources and to consumer preferences. For this project, we invited six experts in various areas (information science, consumer organizations, fair trade, public administration-cooperation office and telecommunication) to participate and fill the questionnaires. The results are analyzed by market experts in terms of comparison and conformity.

**Keywords:** consumer factors; decision-making model; DEMATEL; AHP; BWM; sustainable supply chain

## 1. Introduction

A sustainable product comes from a stable production and procurement system and more specifically, from a sustainable supply chain (SSC). A sustainable production system should guarantee whether the existing sophisticated processes are functioning based on the triple bottom line concept, addressing economic, environmental, and social concerns. According to this concept, these are the three pillars and a wide range of small and large companies establish such strategies in order to reflect a sustainable system. In addition, while a company projects to sustainable structure, it should be ensured the whole product cycle from the initial point like purchase, design, production, distribution, and disposal system, and consumers act economically, sustainably and environmentally is carried out in an effective way [1,2].

SSC was developed as a fundamental product evaluation instrument in order to keep consumers satisfied, manage a constant relationship with them, and deliver an acceptable level of loyalty. Sustainability in SC became a strategic business function as many companies recognize it as a very significant element in customer satisfaction [3,4]. This concern is highlighted in SSC management, and managers believe noticing to consumer expectation is vital for the future of the company [5].

Mostly, the consumer satisfaction is very relevant to the type of operations, functions, and practices the company is handling [6], and in this way, the whole SC operations must be monitored precisely. To control operational activities of an SSC, sort of variables and factors must be determined. To name a few, setting sustainable strategy and policy, having a sustainable design for products, sustainable sourcing, and effective reverse logistic and disposal system are items that convincingly improve the efficiency of the SC. The art of leaders is to formulate such a strategy to evaluate those items optimally. How to deal with this matter in a firm or manufacturing company is a challenge among industrial sectors. In reality, these factors allow a firm to succeed or fail and collapse. Therefore, keeping in mind that organization, evaluation, and control of them includes a critical responsibility that effectively must be addressed through a robust structure.

Problem-solving tools are interpreted in different ways like finding the best or the most preferable element from a set of available options, and seeking the value or measure all parties agree. Problem-solving usually connects to the uncertain conditions that need making an effective decision using a lot of variables. Advanced companies and organizations invest in their decision-making process carefully and control variables and conditions. The multi-criteria decision-making (MCDM) is an optimization process of identifying the best feasible solution according to the predefined criteria (while each criterion seeks different orientation). Algorithmic thinking and model building in MCDM provide a contemporary approach for explaining certain kinds of human behavior and decision-making [7,8]. MCDM is introduced as one of the operational fields of interdisciplinary research in management and business sciences [9]. In each multiple criteria problem, two main parameters are alternatives (options) and criteria. Therefore, MCDM techniques are classified to produce a ranking of the alternatives and to weight the criteria. For example, the technique for order performance by similarity to ideal solution (TOPSIS) is a method to compare alternatives and prioritize them [10,11]. The other methods relevant to this category are Combined Compromise Solution (CoCoSo) [8]. On the other side, there are methods like analytical network process (ANP) [12] to rank decision criteria and obtain the weights. In this paper, as both interaction and interrelationship of decision criteria are considered, decision-making trial and evaluation laboratory (DEMATEL), analytical hierarchy process (AHP) and best worst method (BWM) methods are used as decision analysis tools to evaluate the consumer attributes. The information and outcomes of the first two methods are integrated as inputs to BWM.

To sum up, the paper is organized as follows: Section 2 presents a literature review, historical background, research gap, and impacts. The adopted methodologies including DEMATEL, AHP and BWM and their step by step operations are presented in Section 3. The implementation of the weighing tools in a real time case study along with discussions on results is explained in Section 4.

## 2. Review of the Literature and Contribution

### 2.1. The Studies on Importance of Mobile Phone Selection Factors According to Customer's Attitude

By 2019, the number of mobile phone users is expected to be 4.68 billion, mostly energy-intensive smartphones users (2.7 billion) [13]. Factors such as growing dependence or affordability have contributed to the average global smartphone replacement cycle has reached 21 months. This trend has been accentuated even more among consumers in emerging markets [14]. This high replacement ratio is attributed by the mobile industry to the tendency of consumers to demand a device with more applications. However, contract length and incentive programs are powerful reasons for users to replace their phone, according to research carried out by Motorola Labs and the University of Toronto. These users' behaviors pose, without doubt, environmental challenges. Sustainability in mobile computing is an urgent problem to address [15].

For years, product design and supply chain decisions in the IT sector have been troubled by transitory, profit-driven perspectives and a linear manufacturing model despite the fact that the functionality of the phones has been innovated [16]:

- In order to extract metals for these devices, miners in isolated areas perform life-threatening work, often stimulating armed conflicts in countries like the Democratic Republic of the Congo (DRC) and destroying the land;
- Damage to the health of workers in electronic factories that are exposed to hazardous chemicals without their knowledge;
- Increasing device complexity means greater amounts of energy is required to produce each phone which in turns increases demand for coal and other forms of dirty energy in China and other parts of Asia;
- Insufficient product take-back and reuse of materials further contribute to a rapidly growing e-waste stream.

Several ways to correct the called "vicious cycle of consumption, waste, and injustice" by the use of smartphones in a global society have been proposed. They require taking a hard look at individual and market behaviors in the smartphone industry, as well as an awareness of both the environmental and human impact of one of the most advanced technological products on the market. Bask et al. [17] believe that the moment that consumers would just think of physical features and appearance for a smartphone, factors like the way of supplying materials, labor force treatment and human rights are concerned more. Despite this, a review of the literature on the criteria for selecting mobile phone consumers shows that most of the studies have focused on the analysis of functional/utilitarian factors (e.g., memory, processor, touch panel, operating system, remote control services, location-based services, mobile wallet services, mobile multimedia services, size, weight or feel in one's hand), hedonic/social factors (e.g., aesthetics, novelty, interactivity-social element, credibility and intimacy, personalization), brand equity/brand effect factors, and price and promotions factors [18,19].

This paper adopted a scheme based on four dimensions for the analysis of mobile sustainability issues, including sustainable strategy and policy, sustainable product design, sustainable sourcing and end-of-life-management [17].

Additionally, two relevant indicators, namely working conditions and cooperative efforts have been incorporated according to the literature on socially responsible consumption [20–23] which can influence purchase decisions. Table 1 shows the definition and information of factors and indicators of mobile phone selection.

**Table 1.** The structure of factors and indicators.

| Factors | Indicators | Description |
|---|---|---|
| Sustainable strategy and policy (F1) | $X_1$—A structural sustainable impact-assessment tool is in place | Has a sustainable management system (ISO, EMS, and AA1000), publishes corporate social responsibility report, informs consumers about actions that support sustainable development, participates in global sustainable development initiatives |
| | $X_2$—Working conditions follows common ethical principles | Equal pay for equal work and fair remuneration, health and safety at work, increase the commitment of workers and promote dialogue between workers and management, provide managers with the necessary skills to improve both employment practices and health and safety |
| | $X_3$—Cooperative efforts with a non-profit organization for mutual benefit | Part of the profits obtained with the sale of telephones are delivered for good causes, manufacturer makes donations to good causes |
| Sustainable product design (F2) | $X_4$—Sustainable material usage and preparedness for recycling | Renewable resources, energy efficient as possible, the origin of the pieces that compose it is traced looking for materials that are good for people and for the planet, recycling and disassembly are taken into account |
| | $X_5$—Management of hazardous materials | Imposes more stringent requirements with regard to hazardous materials than regulations demand, uses a third-party certified analytical tool in the product-design phase in reporting this |
| | $X_6$—Extended life-cycle | Design for reliability and robustness, seek for extension of service life by focusing on modularity and ease of repair, possibilities of upgrade, update or modify it according to user's need, mobile phone compatible in the long term, repair cost lower than replacement costs, repair themselves/easy repairs, offers incentives to keep currents phone |

**Table 1.** *Cont.*

| Factors | Indicators | Description |
|---|---|---|
| Sustainable sourcing (F3) | X7—Sustainable Package and delivery management | Sustainability should be taken into account in the selection of transportation mode and materials, the product packaging should be as efficient as possible |
| | X8—Selecting sustainable suppliers | Purchase of materials from mines that empower vulnerable communities or that have better sustainable performance, has a certified tool in use for evaluating the sustainability of suppliers, trains its suppliers, shares information, audits its suppliers, has clear instructions |
| Sustainable end-of-life-management-disposal (F4) | X9—Move towards a circular economy with better recycling of electronic devices | Collection of old telephones for reuse and recycling, encourage the disposal of primary, physical material, persuade consumers that refurbished or second-hand mobile are "cool" |

## 2.2. History of Decision-Making Tools in Application of Mobile Phone Selection

The methods of analysis used in these studies varied. Among them, considerable works stand out the use of MCDM methods [18,24–34]. However, the literature in which they consider sustainability criteria when choosing a mobile phone by consumers is much lower [6,15,17,35–39] and none of them uses MCDM.

The importance of our study lies within several aspects. A considerable amount of studies in the history of MCDM focus on individual methods to gain the weights of factors or variables. The noticeable point is that, to rate and evaluate mobile phone criteria, classical analysis tools are unable to direct us to an optimal solution. Having in mind to generate the importance of those criteria, we require a set of concrete and standard tools to facilitate the ranking measurement. MCDM methods are the most preferred and recognized tools for multi-criteria evaluation under conflicting environment [2,40]. The contribution of this field in decision-making theory is very high and researchers progressively use these tools in a wide range of applications [41,42]. MCDM methods are generally used to rank alternatives as well as to derive the weight of each decision criterion. Thus, this paper endeavors to identify the most effective factors for selecting smartphones because this aids users and customers in order to be able to select an ethical or sustainable phone. In addition, an analytical approach-based on decision-making tools and expert judgments is presented.

Till date, there is no past study available in the literature focussing on mobile or smartphone evaluation. Few studies have been carried out, which is summarized in Table 2. For instance, Mahdavi et al. [33] proposed an approach for optimal selection of phone mobile fitting to the preferences of the users. In this paper, authors used several versions of AHP, Voting AHP, with Entropy and finally TOPSIS to rank mobile phones under the case of telecommunication center. The idea of using customer approach in design and its further selection of mobile phones might be relevant to the use of quality function deployment (QFD). [27] believe that to select mobile phones, due to some difficulty, QFD can aid to involve customer requirements for as input. They developed appropriate technical requirements. Büyüközkan and Güleryüz [26] integrated fuzzy theory to TOPSIS method in order to deal with uncertainty in a process of smart mobile phone selection. Application of analytical network process and generalized Choquet integral (GCI) byYildiz and Ergul [25], offered a mechanism for clients to rely on a selection strategy for their desirable mobile phones. In India, a group of experts modeled a structure for smartphone selection using hardware, economic and physical attributes and multi-criteria tools called valuation based on distance from average solution [43]. The only research we found dedicated to smartphone improvement is the work of Hu, Lu, and Tzeng [18]. The authors have focused on the customers need to enhance the value of the product in several firms. A combined approach of DEMATEL-ANP and VIKOR were assumed. The paper investigates on three dimensions like product function, mobile convenience, and customer quality. However, after a deep survey in the history of smart or mobile phone selection, the literature lacks concrete perspectives on the applications of decision-making methods in ranking the various factors and indicators of an ethical phone.

**Table 2.** Multi-criteria decision-making (MCDM) techniques used in mobile phone selection research.

| Author | MCDM Method | Objective |
|---|---|---|
| Isiklar and Büyüközkan, 2006 [34] | AHP, TOPSIS | Evaluate the mobile phone options with respect to the users' preferences order |
| Mahdavi et al. 2008 [33] | AHP-ENTROPY-TOPSIS | Right selection of phone mobile fitting to the preferences of the users |
| Pigneur, Ondrus and Bui, 2010 [30] | ELECTRE | Assessing the mobile payment market |
| Chen et al. 2012 [31] | AHP | Mobile phone recommendation system for online stores and consumers |
| Akyene, 2012 [29] | Entropy, TOPSIS | Aid customer in selecting which mobile phone to purchase |
| Saket et al. 2014 [27] | QFD | Selection of appropriate mobile to the customers |
| Cerit, Küçükyazici and Kalem, 2014 [28] | QFD | New product development in accordance with customer expectation |
| Hu, Lu and Tzeng, 2014 [18] | DEMATEL-Based ANP, VIKOR | Provide useful information to enterprises regarding how to optimally satisfy customer needs |
| Yildiz and Ergul, 2015 [25] | ANP, GCI | The best smartphone selection for consumers |
| Büyüközkan and Güleryüz, 2016 [26] | IF-TOPSIS | Ranking appropriate mobile phone alternatives for consumers |
| Srivastava et al. 2017 [24] | AHP | Comparison between smartphones on the basis of their reliability factors for consumers |

Some other investigations for selecting mobile phone factors were done but not systematically. Bask, Halme, and Kuula [6] utilized a conjoint-based analysis (CBC) approach for studying sustainability features that affect consumer evaluations in the mobile phone industry. CBC is used for conducting studies over the web and in CAPI (computer-aided personal interview) interviewing mode where the device is not necessarily connected to the internet, or via paper-and-pencil questionnaires. CBC studies are used for learning about respondents' preferences for the combinations of features that make up products or services. Such analysis can help with product design, line extensions, pricing research, and market segmentation [44]. There are also numerous opportunities for using CBC in modeling economics and healthcare choices. In another study, Bask, Halme, Kallio and Kuula [17] employed CBC to identify relevant product features related to sustainable development, and the choice of a mobile phone as an example in measuring their importance.

According to the aforementioned context, evidence and previous studies, this paper attempts to evaluate and analyze the interrelationship between different mobile phone factors based on a sustainable perspective using the three considered techniques. The combination of DEMATEL and AHP, and BWM methods was defined to enhance the efficiency of the study and quality of results. At firstly, the AHP method is applied to identify the priority of each criterion for assessing a sustainable mobile phone. Then, the results of AHP are tested with the outcomes of DEMATEL approach to obtain the final rank of phone selection factors. In the final stage, an LP model was developed using the outputs from both AHP and DEMATEL methods to be utilized as inputs in BWM. The uniqueness of the analytical study is that for the first time, a multiple criteria-based investigation was performed for the evaluation of sustainable criteria for mobile phone users.

## 3. Materials and Methods

To evaluate the importance of mobile phone factors, three MCDM methods were implemented to obtain the results. Initially, the algorithms of DEMATEL and AHP methods were applied and eventually the BWM results were interpreted.

### 3.1. DEMATEL Method

DEMATEL method consists of the following seven steps [45–47]. It presumes a system restraining a set of components (or factors, criteria) $C = \{C_1, C_2, \ldots, C_n\}$, with pair-wise relations that can be assessed. This is a method which can be utilized for estimation of criteria weights. The steps of DEMATEL method is explained as follows:

Step 1: Generation of the direct-relation matrix (*A*) by scores:

At first, the decision maker (DM) indicates the relationship between the sets of paired criteria that signifies the direct effect that each $i_{th}$ criterion exerts on each $j_{th}$ criterion, as specified by an integer score ranging from 0 to 4, representing no influence, low influence, medium influence, high influence and very high influence. As a result of these assessments, a direct-relation matrix (*A*) is obtained in the form of an $n \times n$ matrix, in which the individual element ($a_{ij}$) denotes the degree to which $i^{th}$ criterion affects $j_{th}$ criterion and *n* denotes the total number of criteria.

$$
A = \begin{bmatrix}
0 & a_{12} & \dots & a_{1j} & \dots & a_{1n} \\
a_{21} & 0 & \dots & a_{2j} & \dots & a_{2n} \\
\dots & \dots & \dots & \dots & \dots & \dots \\
\dots & \dots & \dots & \dots & \dots & \dots \\
a_{n1} & a_{n2} & \dots & a_{nj} & \dots & 0
\end{bmatrix} \tag{1}
$$

Step 2: Formation of the normalized direct-relation matrix(*X*):

After the generation of the direct-relation matrix (*A*), the normalized matrix (*X*) is computedby dividing each element by the maximum value of the sum of the columns and rows, as shown by Equation (2). Each element in matrix *X* ranges from 0 to 1.

$$
X = k \times A \tag{2}
$$

where

$$
k = \frac{1}{\max\limits_{1 \le i \le n}\left(\sum\limits_{j=1}^{n} a_{ij}\right)}, \ i, j = 1, 2, \dots, n \tag{3}
$$

Step 3: Computation of the total-relation matrix (*T*):

The total-relation matrix (*T*) is obtained by Equation (5), in which *I* denotes the identity matrix. Each element ($t_{ij}$) of this matrix symbolizes the indirect influences that $i_{th}$ criterion imparts on $j_{th}$ criterion, and the matrix *T* reveals the total relationship between each pair of decision variables.

$$
T = \left[t_{ij}\right]_{m \times n}, i, j = 1, 2, \dots, n \tag{4}
$$

$$
T = X + X^2 + X^3 + \dots + X^k = X(I + X + X^2 + \dots + X^{k-1})[(I - X)((I - X)^{-1}] = \\
X(I - X^k)(I - X)^{-1} \tag{5}
$$

Then,

$$
T = X(1 - X)^{-1}, k \to \infty, \ X^k = [0]_{n \times n} \tag{6}
$$

Step 4: Determination of the sums of rows and columns of matrix *T*:

In the total-relation matrix *T*, the sum of rows and sum of columns are represented by vectors *D* and *R*, as derived using Equations (7) and (8):

$$
D_i = \left[\sum_{j=1}^{n} t_{ij}\right]_{n \times 1} = [t_i]_{n \times 1}, \ i = 1, 2, \dots, n \tag{7}
$$

$$
R_j = \left[\sum_{i=1}^{n} t_{ij}\right]_{1 \times n} = [t_j]_{n \times 1}, \ j = 1, 2, \dots, n \tag{8}
$$

Step 5: Setting a threshold value ($\alpha$):

The obtained matrix $T$ provides information on how one factor affects another. It is completely essential for the decision maker to set a threshold value ($\alpha$) for elucidating the structural relation among criteria while simultaneously keeping the intricacy of the entire system to a convenient level. If their correlation value in matrix T is smaller than $\alpha$, an influence relationship between two elements is excluded from the map. While only the effects greater than the set $\alpha$ value are chosen and shown in the digraph. The value of $\alpha$ is computed using Equation (9), where $N$ is the total number of elements in matrix $T$.

$$\alpha = \frac{\sum\limits_{i=1}^{n} \sum\limits_{j=1}^{n} [t_{ij}]}{N} \tag{9}$$

Step 6: Development of a causal diagram:

The causal diagram illustrates a classification of the degree of each criterion and explains the criterion which can be classed as either a passive one or active one. The horizontal axis vector ($D_k + R_k$) named 'prominence' is computed by adding $D$ to $R$ while $k = i = j = 1$ which reveals how much importance the criterion has and indicates the criterion which affects others and is affected by others. The vector ($D_k + R_k$) denotes the weights of each criterion and we call it ($q_j$). Similarly, the vertical axis ($D_k - R_k$) named 'relation' is obtained by subtracting $D$ from $R$. This allows us to divide the criteria into a *cause* group and an effect group. Generally, when the value of 'relation' is positive, the criterion belongs to the causal group and if the value is negative, the criterion belongs to the effect group. Hence, causal diagrams visualize the complicated relationships and interaction influence levels between the decision criteria into a visible structural model. This function provides a useful insight for problem solving. According to DEMATEL, the DM can realize the driving variables of the core problem in a complicated system, and make suitable decisions to solve the problem with regard to attribute type and influence level.

*3.2. Analytical Hierarchy Process*

One of the classical tools in decision-analysis is AHP, which has been increasingly implemented in various applications. It is recognized as an effective problem-solving technique in environmental and climate changes [48], university ranking [49–51]. The anatomy of AHP lets its users to achieve a ranking of alternatives and also generate the importance weights of decision factors (criteria) as well. AHP stepwise procedure to carry out the relative importance of criteria is represented here. AHP methodology follows the steps below to find relative importance degree of criteria:

Step 1: Develop the pairwise comparison matrix A by utilizing the ratio scale in Table 3.

**Table 3.** Scale ratio for pairwise comparison by experts [52].

| Intensity of Importance | Definition |
|:---:|:---:|
| 1 | Equally important |
| 3 | Moderately important |
| 5 | Strongly more important |
| 7 | Very strong important |
| 9 | Extremely more important |
| 2,4,6,8 | Intermediate more important |

Step 2: Let $C_1, C_2, \ldots\ldots, C_n$ as the set of elements, although $a_{ij}$ presents a quantified judgment on pair of elements $C_i, C_j$ the matrix A as below:

$$A = \left[a_{ij}\right] = \begin{bmatrix} 1 & a_{12} & \dots & a_{1n} \\ \frac{1}{a_{12}} & 1 & \dots & a_{2n} \\ \vdots & \vdots & \vdots & \vdots \\ \frac{1}{a_{1n}} & \frac{1}{a_{2n}} & \dots & 1 \end{bmatrix} \tag{10}$$

where, $a_{ij} = 1$ and $a_{ji} = \frac{1}{a_{ij}}, i, j = 1, 2, \dots, n$

Step 3: In matrix A, the problem is on determining a set of numerical weights $W_1, W_2, \dots, W_n$ in front of $n$ element $C_1, C_2, \dots, C_n$. Relation between the weights $W_i$ and judgments $a_{ij}$ are given by $a_{ij} = \frac{W_i}{W_j}$, for $(i, j = 1, 2, 3, \dots, n)$. Matrix A is said to be consistent if $a_{ij} \times a_{jk} = a_{ik}$ and its principal (largest) eigen value ($\lambda_{max}$) is equal to $n$, or in other words, $a_{ij} = \frac{W_i}{W_j}$ holds if and only if consistency ratio (*CR*) = 0. $\lambda_{\max}$ is estimated as [52]:

$$\lambda_{\max} = \sum_{j=1}^{n} a_{ij} \frac{W_i}{W_j} \tag{11}$$

In AHP, it is important to meet the following condition for consistent results:

$$(A - \lambda_{\max} I)X = 0 \tag{12}$$

Step 4: Consistency index (*CI*) is estimated as:

$$CI = \frac{\lambda_{max} - n}{n - 1} \tag{13}$$

In AHP, *CR* is obtained using $CR = \frac{CI}{RI}$, where *RI* is the random index. The number 0.1 is the accepted upper limit of *CR*. When *CR* > 0.10, the evaluation process of pairwise comparison should be repeated to improve consistency.

### 3.3. Best Worst Method

Rezaei [41] developed BWM as an MCDM technique based on a linear programming perspective and received considerable attention in various fields [53,54]. This method directs the decision-making problem in order to find the weight and rank of decision criteria. The idea behind the BWM allows decision makers to run an operable model in complex decision environments [55]. Wide range of applications adopted the method in order to find an optimal solution [56–58]. The steps below are the process to obtain weights of decision criteria:

Step 1: The decision maker (DM) determines a set of decision criteria: $\{c_1, c_2, \dots, c_n\}$

Step 2: The DM chooses the best and the worst criteria. In this step, the DM chooses the best and the worst criteria among the set of identified criteria in last step. The best criterion represents the most desirable or the most significant one, while the worst criterion is the least important one among others.

Step 3: The DM conducts pairwise comparisons between the best criterion and the other criteria. In this step, the goal is to identify the preference of the most important criterion to the other criteria. DM uses a scale from 1 to 9 (1: equally important, and 9: extremely more important). The comparison outcome is described as best to another vector: $A_B = (a_{B1}, a_{B2}, \dots, a_{Bn}$, where $a_{Bj}$ represents the preference of the best criterion B over the criterion $j$ and $a_{BB} = 1$

Step 4: The DM conducts a pairwise comparison between the other criteria and the worst criterion. The same as last step, the comparison results are expressed by other-to-worst vector: $A_W = (a_{1W}, a_{2W}, \dots, a_{nB})^T$ where $a_{jw}$ represents the preference of the best criterion $j$ over the criterion $W$ and $a_{WW} = 1$

Step 5: Calculating the optimal weights: $\left(W_1^*, W_2^*, \ldots, W_n^*\right)$

For each pair of $\frac{W_B}{W_j}$ and $\frac{W_j}{W_W}$, the optimal weight should meet the requirement that $\frac{W_B}{W_j} = a_{Bj}$ and $\frac{W_j}{W_W} = a_{jW}$. To satisfy the conditions, the maximum absolute differences $\left|\frac{W_B}{W_j} - a_{Bj}\right|$ and $\left|\frac{W_j}{W_W} - a_{jW}\right|$ for all $j$ is minimized. Moreover, taking into consideration the non-negativity characteristic and sum condition of the weights, the following problem can be formulated:

$$\text{Min} \max_j \left\{ \left|\frac{W_B}{W_j} - a_{Bj}\right|, \left|\frac{W_j}{W_W} - a_{jW}\right| \right\} \tag{14}$$

Subject to

$$\sum_j W_j = 1, \ W_j \geq 0 \ \text{for all } j \tag{15}$$

The model can be transformed as:

$\min \xi$,

Subject to

$$\left|\frac{W_B}{W_j} - a_{Bj}\right| \leq \xi, \ \text{for all } j \tag{16}$$

$$\left|\frac{W_j}{W_W} - a_{jW}\right| \leq \xi, \ \text{for all } j \tag{17}$$

$$\sum_j W_j = 1, \ W_j \geq 0 \ \text{for all } j \tag{18}$$

After finding the results, we should calculate the consistency level of the comparisons. The *CR* of BWM can be expressed by using $\xi^*$ and the corresponding *CI* (Table 4), as follows:

**Table 4.** *CI* of BWM.

| $a_{BW}$ | 1 | 2 | 3 | 4 | 5 | 6 | 7 | 8 | 9 |
|---|---|---|---|---|---|---|---|---|---|
| *CI* | 0.00 | 0.44 | 1.00 | 1.63 | 2.30 | 3.00 | 3.73 | 4.47 | 5.23 |
| $CR = \frac{\xi^*}{CI}$ | (19) | | | | | | | | |

It can be seen that the smaller the $\xi^*$, the smaller the *CR* value, and the more consistent the vectors are. Statistical analysis of Rezaei [41] has established that BWM accomplishes significantly better results than AHP with respect to the *CR*, minimum violation (preservative ability of ordinal preferences), total deviation (actual Euclidean distance between the ratios of any weights of any criteria pair and their corresponding pairwise comparisons) and conformity (intuitive evaluations by the DMs).

## 4. Case Definition, Model Implementation, and Results

### 4.1. Case of Mobile Phone Sustainable Factors Evaluation

An MCDM approach was designed in order to analyze the significance of the sustainable factors for a mobile phone case study. The process of evaluating the factors is summarized in Figure 1.

Step 1: Identifying the influential factors and indicators (criteria) with respect to sustainability preferences of the consumers. In addition, as a quantitative analysis of the factors will be performed, experts in various areas are invited to participate and fill a set of developed questionnaire. Respondents (experts) are professionals from a fair trade association, an association of consumers, public administration,

IT production, and communication sector. Three experts are from Spain and the other three are from India. The first expert is from an association of consumers, who have had the role of President of CECU (Spanish Association of Users and Consumers), and currently is a member of several advisory councils, including the Spanish CSR Observatory and the Journal "Corresponsables". The second expert is a technician from a public administration-cooperation office and graduated from engineering school with years of experience in this area. The third person is an expert and General Director of Fairtrade Ibérica with a postgraduate degree. The three respondents from India are the experts of electronics consumer sector and have several years of experience in mobile phone industries. The designed questionnaires were sent and distributed among the experts. The experts were requested to rate the importance of factors and indicators on a scale of 0 to 4. Afterwards, they make pairwise comparison and offer their opinion as to if they are comparing the influence of each factor (indicator) over another one.

Step 2: The obtained data are aggregated by taking the average of the experts' opinions in order to use in subsequent steps. It was assumed to use the same importance of the experts in the entire decision-making process.

Step 3: In order to know the importance of consumer factors, DEMATEL and AHP methods are implemented. DEMATEL is a method that categorizes the cause and effect group of the factors and generates the ranking of the factors along with their weights, while AHP compares criteria pairwise and estimates the importance of each decision criteria.

Step 4: Identifying the best and worst factors based on the results of DEMATEL and AHP to subsequently use as inputs to BWM to produce the aggregated weights.

Step 5: Discussing and analyzing the factors and giving feedback to the experts.

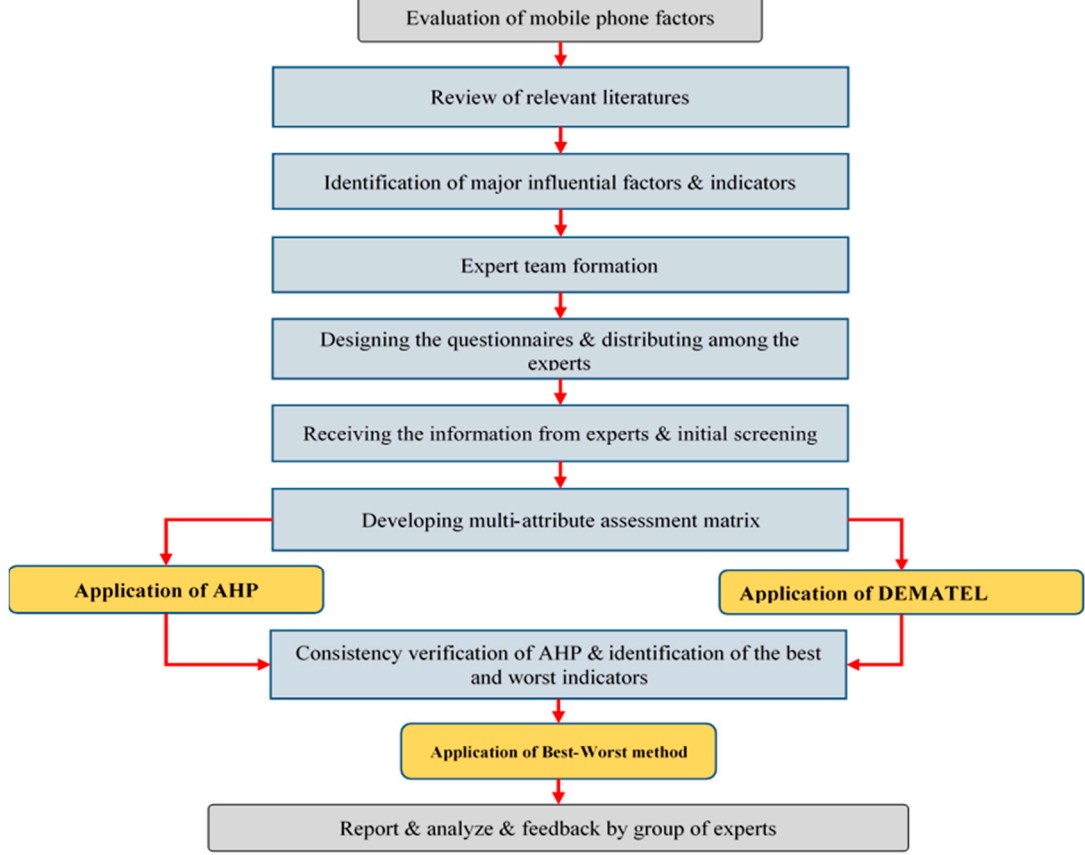

**Figure 1.** The step by step procedure for mobile factors weights formulation.

## 4.2. Results and Discussion

### 4.2.1. DEMATEL Implementation

Since evaluation of the sustainable smartphone is a very intricate and complex multi-criteria problem, it is inappropriate to presume its different components to be independent. As all of the nine identified indicators (sub-factors) are perceptibly important, hence, it becomes indispensable to find out the importance of the sub-factors for this assessment and quantify their relationships. To accomplish this, the DEMATEL method is used for encapsulating the inter-relationships between those sub-factors. Pursuing the methodological procedures of DEMATEL as presented in Section 3.1, the relationships between different factors ($F_1$ to $F_4$) and indicators ($X_1$ to $X_9$) are scored by the experts using the previously mentioned integer scale and six filled questionnaires. Once these relationships are quantified, the initial direct-relation matrix ($A$) is developed based on group decision agreement using arithmetic mean of all individual opinions, as shown in Table A1 of Appendix A. It is a $9 \times 9$ matrix, obtained by pair-wise comparisons in terms of influences and directions between the considered indicators. In this paper, DEMATEL method performs two-level computations namely for the indicators and the main factors. The obtained weights of the factors as computed using DEMATEL are found as $w_{f1} = 0.25$, $w_{f2} = 0.245$, $w_{f3} = 0.275$, $w_{f1} = 0.245$. Due to page restrictions, the detail process of computing the weights of the indicators is not presented here. The normalized matrix is obtained from Table A1 of Appendix A using Equations (2) and (3) respectively, as presented in Table A2 of Appendix A. Then, the total-influence matrix ($T$) is calculated using Equation (5), as shown in Table A3 of Appendix A. Now, the sum of each row and column results in two vectors named 'prominence' ($D + R$) and 'relation' ($D - R$) respectively, employing Equations (7) and (8) respectively, as shown in Table 5. The information provided in columns ($D + R$) and ($D - R$) in Table 5 indicate the degree of total influence levels and the degree of net influence levels respectively. The positive values indicate that it will influence other indicators more than any other indicator influences it. Now, looking at the ($D + R$) and ($D - R$) values of Table 5, it is completely comprehensible that the nine indicators are divided into cause and effect groups. The cause group consists of five indicators, i.e.; $X_1$, $X_2$, $X_3$, $X_6$, and $X_7$ and the effect group contains the remaining three criteria ($X_4$, $X_5$, $X_8$, and $X_9$). It is obvious that indicators $X_1$, $X_2$, $X_3$, $X_6$, and $X_7$ are the main driving elements for $X_4$, $X_5$, $X_8$, and $X_9$. Among these nine sub-factors, $X_1$ (structural sustainable impact-assessment tool) is recognized as the most significant one because it has the maximum intensity of relation to others for having maximum $D + R$ value followed by $X_3$ (cooperative efforts with a non-profit organization for mutual benefit). Thus, $X_1$ and $X_3$ play major roles in the evaluation problem, having the maximum impact on the others. Moreover, Table 5 also signifies that working conditions and ethical principles ($X_2$) is the least important indicator among all, having the least $D + R$ value. On the other hand, $X_8$ (selecting sustainable suppliers) is very much influenced by the other sub-factors, having the lowest ($D - R$) value. Local weights of the indicators are calculated by normalizing the values of prominence vector $D + R$, as shown in Table 5. The global weights of the indicators (sub-factors) are computed by normalization of the multiplied value of the local weights of the indicators and local weights of the factors. From the values of Table 5, it is observed that as $X_1$ is the most influencing indicator, and it has the highest weight among other indicators. Figure 2 shows the cause and effect diagram for interrelationship among the nine indicators. This diagram is designed using Equations (7) and (8) respectively and according to the data presented in Table 5.

**Table 5.** Decision-making trial and evaluation laboratory (DEMATEL) computations and weights of sub-factors (indicators) for sustainable mobile phone assessment.

| Factors with Local Weight (FLW) | Indicator | $D + R$ | $D - R$ | Local Weight of Indicator (ILW) | (FLW) × (ILW) | Normalized Global Weight | Group | Rank |
|---|---|---|---|---|---|---|---|---|
| | $X_1$ | 5.1797 | 0.1524 | 0.1410 | 0.0353 | 0.1414 | Cause | 1 |
| $F_1$ (0.25) | $X_2$ | 3.0821 | 0.4612 | 0.0840 | 0.0210 | 0.0841 | Cause | 9 |
| | $X_3$ | 4.8867 | 0.8596 | 0.1330 | 0.0333 | 0.1334 | Cause | 2 |

**Table 5.** *Cont.*

| Factors with Local Weight (FLW) | Indicator | $D + R$ | $D - R$ | Local Weight of Indicator (ILW) | (FLW) × (ILW) | Normalized Global Weight | Group | Rank |
|---|---|---|---|---|---|---|---|---|
| F_2 (0.245) | X_4 | 4.0320 | −0.9052 | 0.1100 | 0.0269 | 0.1079 | Effect | 5 |
| | X_5 | 3.3757 | −0.0156 | 0.0920 | 0.0225 | 0.0903 | Effect | 7 |
| | X_6 | 3.6954 | 0.8862 | 0.1010 | 0.0247 | 0.0989 | Cause | 6 |
| F_3 (0.275) | X_7 | 3.1153 | 0.2239 | 0.0850 | 0.0221 | 0.0884 | Cause | 8 |
| | X_8 | 4.8423 | −1.1276 | 0.1320 | 0.0343 | 0.1375 | Effect | 3 |
| F_4 (0.245) | X_9 | 4.3979 | −0.5348 | 0.1200 | 0.0294 | 0.1177 | Effect | 4 |

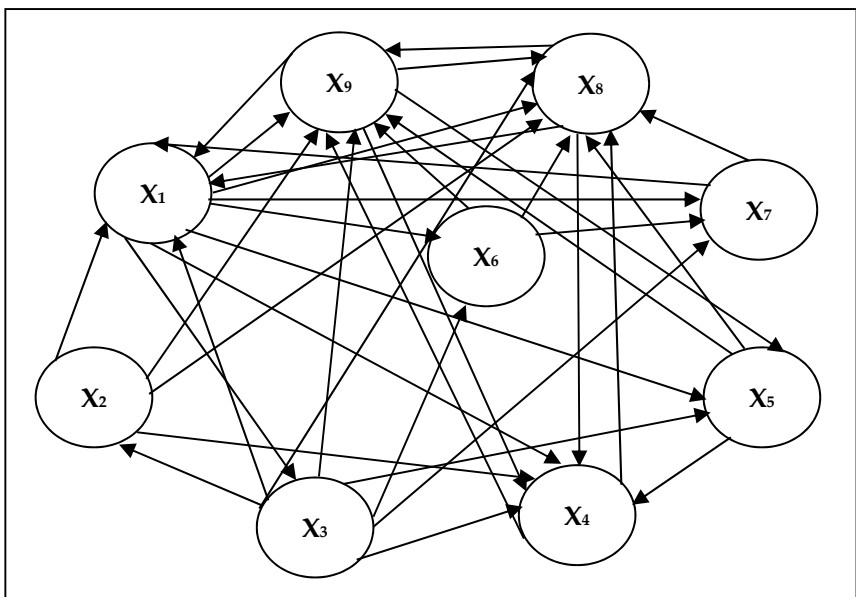

**Figure 2.** DEMATEL cause and effect diagram.

### 4.2.2. Implementation of AHP

This section solves the problem of finding the most crucial indicators for mobile phone selection according to the AHP method. Similar to the DEMATEL model, experts were requested to compare the indicators pairwise. The first matrix (*A*), generated by the experts after pairwise comparisons, is shown in Table 6. Based on the matrix *A*, a normalized matrix is computed by dividing the elements of each column by the sum of the same column. Sum of each row of the normalized matrix indicates the local weights ($W_j$), as given by AHP. All these values are given in Table A4 of Appendix A. The multiplication of matrix *A* and vector $W_j$ gives the first column of Table A5. After that, $(A) \times (W_j)/W_i$ gives the values of $\lambda_{max}$, as shown by Equation (11). It is important to meet the condition of Equation in order to achieve a consistent result using AHP, as shown in Table A5 of Appendix A. Now, using Equation (13), *CI* value is estimated as 0.1401 and the relevant *RI* for 9 criteria (n = 9) is 1.46 [59], therefore, the *CR* value becomes 0.09 which is lower than 0.1. It indicates the judgments made by the experts are acceptably consistent. Table A5 of Appendix A presents the analytical results of AHP consistency check process. Based on the values of Table A4, the global weights of the indicators and their ranks are shown in Table 7. Similar to the results of DEMATEL method, $X_1$ and $X_2$ seem to be the most important and the least important mobile phone indicators.

**Table 6.** Pairwise comparison matrix for analytical hierarchy process (AHP) method.

| Indicator | $X_1$ | $X_2$ | $X_3$ | $X_4$ | $X_5$ | $X_6$ | $X_7$ | $X_8$ | $X_9$ |
|---|---|---|---|---|---|---|---|---|---|
| $X_1$ | 1 | 9 | 2 | 6 | 2 | 3 | 5 | 2 | 4 |
| $X_2$ | 1/9 | 1 | 1/4 | 1/8 | 1/9 | 1/5 | 1/5 | 1/9 | 1/4 |
| $X_3$ | 1/2 | 4 | 1 | 7 | 3 | 2 | 3 | 2 | 3 |
| $X_4$ | 1/6 | 8 | 1/7 | 1 | 1/7 | 1/5 | 1/2 | 1/4 | 1/3 |
| $X_5$ | 1/2 | 9 | 1/3 | 7 | 1 | 2 | 3 | 3 | 2 |
| $X_6$ | 1/3 | 5 | 1/2 | 5 | 1/2 | 1 | 2 | 4 | 2 |
| $X_7$ | 1/5 | 5 | 1/3 | 2 | 1/3 | 1/2 | 1 | 0,5 | 3 |
| $X_8$ | 1/2 | 9 | 1/2 | 4 | 1/3 | 1/4 | 2 | 1 | 2 |
| $X_9$ | 1/4 | 4 | 1/3 | 3 | 1/2 | 1/2 | 1/3 | 1/2 | 1 |
| Sum | 3.5611 | 54.091 | 5.3929 | 35.268 | 7.9167 | 9.65 | 17.033 | 13.361 | 17.583 |

**Table 7.** AHP weights and ranking of the indicators.

| Factors with Local Weight (FLW) | Sub-Factor | $(FLW) \times (W_j)$ | Global Weight | Ranking |
|---|---|---|---|---|
| $F_1$ (0.3319) | $X_1$ | 0.08196 | 0.24694 | 1 |
| | $X_2$ | 0.00621 | 0.01872 | 9 |
| | $X_3$ | 0.06199 | 0.18677 | 2 |
| $F_2$ (0.1949) | $X_4$ | 0.01309 | 0.03944 | 8 |
| | $X_5$ | 0.05239 | 0.15787 | 3 |
| | $X_6$ | 0.04122 | 0.12421 | 4 |
| $F_3$ (0.29) | $X_7$ | 0.02315 | 0.06975 | 6 |
| | $X_8$ | 0.03271 | 0.09855 | 5 |
| $F_4$ (0.1832) | $X_9$ | 0.01917 | 0.05776 | 7 |

### 4.2.3. Implementation of the Best-Worst Method

Based on the results obtained from AHP and DEMATEL methods, it is evident that $X_1$ and $X_2$ are the best (most important) and the worst (least important) indicators respectively for this mobile phone assessment problem. Now, in order to initiate the application of BWM method, aggregated expert opinion regarding the preferences of the most important indicator ($X_1$) over all the other indicators and the preferences of all other indicators over the least important indicator ($X_2$) from Table 6 is again used, as listed in Table 8. The LP formulation based on Equation (14) is then developed, as shown in Box 1. The developed LP model is then the solved model in LINDO and the results are exhibited in Table 9. For our study, *CR* value is found to be 0.0719, as shown in Table 9. Table 9 indicates that among the nine sub-factors or indicators, $X_1$ becomes the most prominent one, whereas $X_2$ is the least important sub-factor.

**Table 8.** Aggregated comparison of the best and worst criteria to other criteria based on AHP pairwise comparison.

| Best to Others | $X_1$ | $X_2$ | $X_3$ | $X_4$ | $X_5$ | $X_6$ | $X_7$ | $X_8$ | $X_9$ |
|---|---|---|---|---|---|---|---|---|---|
| $X_1$ | 1 | 9 | 2 | 6 | 2 | 3 | 5 | 2 | 4 |
| **Others to the Worst** | $X_1$ | $X_2$ | $X_3$ | $X_4$ | $X_5$ | $X_6$ | $X_7$ | $X_8$ | $X_9$ |
| $X_2$ | 1/9 | 1 | 1/4 | 1/8 | 1/9 | 1/5 | 1/5 | 1/9 | 1/4 |

**Box 1.** Developed LP formulation for mobile phone indicators.

$$
\begin{aligned}
&\min \xi, \\
&\text{subject to} \\
&\left|\frac{W_1}{W_2} - 9\right| \le \xi, \left|\frac{W_1}{W_3} - 2\right| \le \xi, \left|\frac{W_1}{W_4} - 6\right| \le \xi, \left|\frac{W_1}{W_5} - 2\right| \le \xi, \left|\frac{W_1}{W_6} - 3\right| \le \xi, \left|\frac{W_1}{W_7} - 5\right| \le \xi, \left|\frac{W_1}{W_8} - 2\right| \le \xi, \left|\frac{W_1}{W_9} - 4\right| \le \\
&\xi \left|\frac{W_2}{W_1} - 9\right| \le \xi, \left|\frac{W_2}{W_3} - 4\right| \le \xi, \left|\frac{W_2}{W_4} - 8\right| \le \xi, \left|\frac{W_2}{W_5} - 9\right| \le \xi, \left|\frac{W_2}{W_6} - 5\right| \le \xi, \left|\frac{W_2}{W_7} - 5\right| \le \xi, \left|\frac{W_2}{W_8} - 9\right| \le \xi; \left|\frac{W_2}{W_9} - 4\right| \le \xi \\
&\sum_{j=1}^{9} W_j = 1 \; W_j \ge 0 \text{ for all } j
\end{aligned}
$$

**Table 9.** Model solution and results of BWM.

| Weights | $X_1$ | $X_2$ | $X_3$ | $X_4$ | $X_5$ | $X_6$ | $X_7$ | $X_8$ | $X_9$ |
|---|---|---|---|---|---|---|---|---|---|
| | 0.231 | 0.027 | 0.152 | 0.051 | 0.152 | 0.101 | 0.061 | 0.152 | 0.076 |
| *(CR)* | 0.0719 | | | | | | | | |

## 5. Conclusions

It is a very well accepted fact that criteria weights play major roles for solving decision-making problems and have a vital contribution for obtaining pragmatic results. In this paper, three different approaches, namely DEMATEL, AHP and BWM methods with quite dissimilar methodological structures were adopted for computation of the weights of the indicators and factors for an ethical smartphone. There is no study in the literature to date with a meaningful mixture of these three tools. Therefore, DEMATEL and AHP are jointly used to initially identify the ranking of the sustainable phone criteria, which are subsequently used for BWM-based analysis. It was found that based on the experts' opinion, the best and worst indicators in AHP and DEMATEL are the same. This is a reason to trust the accuracy of the decision-making process and advance the rest of the process. Then BWM uses the best and worst criteria information (produced by DEMATEL-AHP) in order to derive the optimal weights of the indicators and factors. The choice of criteria and deliberate hierarchy, additionally with expert judgment to determine the level of significance of every criterion on weighting methods is very influential for taking appropriate decisions. Based on the comparison of the three adopted methods, $X_1$ (a structural sustainable impact-assessment tool is in place) received the maximum weights, while working conditions follow common ethical principles ($X_2$) seems to be the least important criterion. However, for the rest of the criteria, these methods do not have a similar ranking sequence in terms of their importance. The BWM method makes the comparisons in a more structured way, which makes it easier and more understandable, and leads to more consistent comparisons, hence more reliable weights can be achieved. One of the main concerns of the AHP refers to the inconsistency of in pairwise comparisons, while the main disadvantage of the DEMATEL method is the lack of consistency measure, i.e.; its inability to validate the results obtained. However, BWM needs substantially less pairwise comparisons, so the chances of inconsistency are thereby reduced, and the *CR* of BWM is used to cross-verify the reliability of the comparisons. The impact of the research also relies on the appropriate participation of experts from Spain and India in related sectors. The analytical results show that structural sustainable impact-assessment tool seems to be the best indicator for a mobile phone sustainability evaluation, and working conditions following common ethical principles appeared as the worst indicator. Thus, the results seem to indicate that consumers in the evaluation of mobile phones sustainability do not consider social criteria very much. The adopted methodology resulted in reduced risk of imprecise judgments and increased quality and reliability of the entire decision-making process.

**Author Contributions:** Conceptualization, M.Y. and M.J.M.-S.; introduction, P.C. and M.Y.; literature review, M.Y. and P.C.; writing—original draft preparation, M.Y., P.C. and M.J.M.-S.; writing—review and editing, R.A.A.-P.; materials and methods, M.Y. and P.C.; Case Definition, Model Implementation, and Results, M.Y., P.C., M.J.M.-S. and R.A.A.-P.

**Funding:** This research received no external funding.

**Conflicts of Interest:** The authors declare no conflict of interest.

**Appendix A**

**Table A1.** Initial direct-relation matrix based on group decision agreement for DEMATEL method.

| Indicator | $X_1$ | $X_2$ | $X_3$ | $X_4$ | $X_5$ | $X_6$ | $X_7$ | $X_8$ | $X_9$ |
|---|---|---|---|---|---|---|---|---|---|
| $X_1$ | 0 | 2 | 3.33 | 3.89 | 2.67 | 4 | 3.67 | 4 | 1 |
| $X_2$ | 4 | 0 | 3 | 3.33 | 0 | 0 | 0 | 1 | 3.67 |
| $X_3$ | 3 | 2 | 0 | 4 | 4 | 3 | 3 | 4 | 4 |
| $X_4$ | 2 | 1 | 2 | 0 | 1 | 0 | 0 | 3.67 | 4 |
| $X_5$ | 2 | 3.33 | 1.89 | 3 | 0 | 0 | 0 | 4 | 1 |
| $X_6$ | 4 | 0 | 2.67 | 3 | 0 | 0 | 4 | 4 | 2.67 |
| $X_7$ | 3 | 0 | 2.67 | 0 | 0 | 3 | 0 | 3 | 2 |
| $X_8$ | 1.83 | 1 | 2 | 1.9 | 2 | 2 | 2 | 0 | 3.89 |
| $X_9$ | 4 | 1.83 | 1 | 3.23 | 3.89 | 0 | 0 | 3.67 | 0 |

**Table A2.** Normalized direct relation matrix of DEMATEL method.

| Indicator | $X_1$ | $X_2$ | $X_3$ | $X_4$ | $X_5$ | $X_6$ | $X_7$ | $X_8$ | $X_9$ |
|---|---|---|---|---|---|---|---|---|---|
| $X_1$ | 0 | 0.0732 | 0.1218 | 0.1423 | 0.0977 | 0.1463 | 0.1342 | 0.1463 | 0.0366 |
| $X_2$ | 0.1463 | 0 | 0.1097 | 0.1218 | 0 | 0 | 0 | 0.0366 | 0.1342 |
| $X_3$ | 0.1097 | 0.0732 | 0 | 0.1463 | 0.1463 | 0.1097 | 0.1097 | 0.1463 | 0.1463 |
| $X_4$ | 0.0732 | 0.0366 | 0.0732 | 0 | 0.0366 | 0 | 0 | 0.1342 | 0.1463 |
| $X_5$ | 0.0732 | 0.1218 | 0.0691 | 0.1097 | 0 | 0 | 0 | 0.1463 | 0.0366 |
| $X_6$ | 0 | 0 | 0 | 0 | 0 | 0 | 0.146 | 0.146 | 0.098 |
| $X_7$ | 0.11 | 0 | 0.098 | 0 | 0 | 0.11 | 0 | 0.11 | 0.073 |
| $X_8$ | 0.07 | 0.037 | 0.073 | 0.0690 | 0.0730 | 0.073 | 0.073 | 0 | 0.142 |
| $X_9$ | 0.146 | 0.067 | 0.037 | 0.1180 | 0.1420 | 0 | 0 | 0.134 | 0 |

**Table A3.** Total direct relation matrix (*T*) for DEMATEL method application.

| Indicator | $X_1$ | $X_2$ | $X_3$ | $X_4$ | $X_5$ | $X_6$ | $X_7$ | $X_8$ | $X_9$ | D |
|---|---|---|---|---|---|---|---|---|---|---|
| $X_1$ | 0.2514 | 0.1945 | 0.315 | 0.371 | 0.2550 | 0.2796 | 0.2752 | 0.4335 | 0.2907 | 2.666 |
| $X_2$ | 0.2998 | 0.0938 | 0.2339 | 0.2863 | 0.1327 | 0.0978 | 0.0978 | 0.2399 | 0.2896 | 1.772 |
| $X_3$ | 0.37 | 0.2128 | 0.2165 | 0.3977 | 0.3188 | 0.2489 | 0.253 | 0.4575 | 0.3978 | 2.873 |
| $X_4$ | 0.2149 | 0.1193 | 0.1821 | 0.1514 | 0.1515 | 0.0823 | 0.0827 | 0.298 | 0.281 | 1.563 |
| $X_5$ | 0.2237 | 0.2004 | 0.1925 | 0.2616 | 0.1114 | 0.0865 | 0.0869 | 0.3152 | 0.2018 | 1.68 |
| $X_6$ | 0 | 0 | 0 | 0 | 0 | 0 | 0.271 | 0.394 | 0.304 | 2.291 |
| $X_7$ | 0.26 | 0.081 | 0.219 | 0.161 | 0.1170 | 0.205 | 0 | 0.294 | 0.223 | 1.67 |
| $X_8$ | 0.241 | 0.13 | 0.207 | 0.239 | 0.1940 | 0.165 | 0.168 | 0 | 0.3 | 1.857 |
| $X_9$ | 0.306 | 0.169 | 0.183 | 0.294 | 0.2600 | 0.101 | 0.101 | 0.339 | 0 | 1.932 |
| R | 2.5137 | 1.3105 | 2.0135 | 2.4686 | 1.695 | 1.404 | 1.4457 | 2.985 | 2.466 | |

**Table A4.** Normalized matrix with local weights of the indicators for AHP.

| Indicator | $X_1$ | $X_2$ | $X_3$ | $X_4$ | $X_5$ | $X_6$ | $X_7$ | $X_8$ | $X_9$ | Local Weight ($W_j$) |
|---|---|---|---|---|---|---|---|---|---|---|
| $X_1$ | 0.281 | 0.166 | 0.371 | 0.17 | 0.253 | 0.311 | 0.294 | 0.15 | 0.227 | 0.247 |
| $X_2$ | 0.031 | 0.018 | 0.046 | 0.004 | 0.014 | 0.021 | 0.012 | 0.008 | 0.014 | 0.019 |
| $X_3$ | 0.14 | 0.074 | 0.185 | 0.198 | 0.379 | 0.207 | 0.176 | 0.150 | 0.171 | 0.187 |
| $X_4$ | 0.047 | 0.148 | 0.026 | 0.028 | 0.018 | 0.021 | 0.029 | 0.019 | 0.019 | 0.039 |
| $X_5$ | 0.14 | 0.168 | 0.062 | 0.203 | 0.126 | 0.207 | 0.176 | 0.225 | 0.114 | 0.158 |
| $X_6$ | 0.094 | 0.092 | 0.093 | 0.142 | 0.063 | 0.104 | 0.117 | 0.299 | 0.114 | 0.124 |
| $X_7$ | 0.056 | 0.092 | 0.062 | 0.057 | 0.042 | 0.052 | 0.059 | 0.037 | 0.171 | 0.07 |
| $X_8$ | 0.14 | 0.166 | 0.093 | 0.113 | 0.042 | 0.026 | 0.117 | 0.075 | 0.114 | 0.099 |
| $X_9$ | 0.07 | 0.074 | 0.062 | 0.085 | 0.063 | 0.052 | 0.02 | 0.037 | 0.057 | 0.058 |

**Table A5.** Calculations for consistency check in AHP.

| (A) × (W$_j$) | (A) × (W$_j$)/W$_i$ | $\lambda_{max}$ | $(A-\lambda_{max}I)X$ | CI | RI | CR |
|---|---|---|---|---|---|---|
| 2.491 | 10.087 | | | | | |
| 0.179 | 9.579 | | | | | |
| 1.963 | 10.51 | | | | | |
| 0.383 | 9.704 | | | | | |
| 1.664 | 10.543 | 10.120 | 0 | 0.140 | 1.460 | 0.090 |
| 1.319 | 10.618 | | | | | |
| 0.691 | 9.909 | | | | | |
| 0.98 | 9.948 | | | | | |
| 0.588 | 10.189 | | | | | |

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
