# Peer review of "An Integrated Multi-Attribute Model for Evaluation of Sustainable Mobile Phone"

_sustainability, doi:10.3390/su11133704_

Round 1
Reviewer 1 Report
no comments
Author Response
Thank you for your considerraqtion. We have submitted the revised version and response to each reviewer as well.
Reviewer 2 Report
The article titled “An Integrated Multi Attribute Model for Mobile 2 Phone Sustainable Criteria Evaluation” proposes a hybrid multi-criteria approach within the field of marketing decision-making. In particular, the authors analyse important factors impacting on customers’ satisfaction when using a phone with sustainable attributes. With this regard, results derived from the application of two different techniques - that are the Analytic Hierarchy Process (AHP) and the Decision-Making Trial and Evaluation Laboratory (DEMATEL) - are compared and the final ranking of factors is lastly achieved by means of the Best Worst Method (BWM).
The article is interesting, presents a well-developed case study and I recommend the publication, being the faced topic within the scope of the Sustainability Journal. The paper presents a pretty clear structure and the main objectives of the study are well formulated. Furthermore, a decision-making team made of six experts with various professional backgrounds has been involved to carry out the study, what I think is particularly useful to collect judgments synthetizing different perspectives. Lastly, the research is supported by a wide list of literature review, citing articles coherent with the main theme of the work.
However, authors should consider and integrate the following suggestions within the article.
- Lines 70/72. After providing a brief description about outcomes of multi-criteria decision-making (MCDM) applications, the authors state that they are going to apply AHP and DEMATEL and eventually the BWM. I would like authors added details about the reasons related to the choice of these particular methodologies. This is clearer in section 2.2 but authors should specify with a higher degree of precision in the introduction why they believe these methods are suitable to treat the field of analysis. With a special reference to the DEMATEL application, the authors may like to check the following paper: Carpitella, S., Carpitella, F., Certa, A., Benítez, J., Izquierdo, J. Managing human factors to reduce organisational risk in industry. Mathematical and Computational Applications, 23 (4) (2018), 67, DOI: 10.3390/mca23040067.
- Section 4.1 starts by describing the involved team of experts. It remains not clear to me if decision-makers have assigned all the same weights to get the aggregated pairwise comparison matrix (Table 10) for the AHP application. I recommend to specify how opinions were collected/aggregated and if experts have the same importance in the decision-making process or not, properly justifying the choice.
- In section 4.2 of results and discussion there are many tables in sequence reporting the detailed development of the various steps of the DEMATEL. I suggest to create an appendix and move there Tables 5,6,7 while keep in section 4.2 just one final table of results, integrating data of tables 8 and 9 into a single table. In this way the whole section would acquire a better look, broadly improving readability. Please, adjust also each caption of tables and check they are correctly referenced in the text.
- A part from the final ranking of factors, the output of the DEMATEL procedure is a causal diagram to graphically show interdependencies among elements (step 6, described in lines 232/244). Please add this chart to section 4.2 and comment results with relation to the specific position of factors in the quadrants of the chart.
- Minor format improvements.
1. The words of line 91 have to be separated.
2. The size of formulas 2/7 (from line 211 to 232) has to be reduced to be uniformed with formulas 9/11 (lines 261/264). Please do the same for formulas 12 and 13.
3. Table 4 has to be separated with spaces from the text above and below (lines 301/303).
4. The general use of English has to be polished and carefully checked out throughout the whole manuscript.
Author Response
Many thanks for reviewing our paper. Below is the detailed resoponses to all your comments.
Sl. No. | Query | Response |
1. | Lines 70/72. After providing a brief description about outcomes of multi-criteria decision-making (MCDM) applications, the authors state that they are going to apply AHP and DEMATEL and eventually the BWM. I would like authors added details about the reasons related to the choice of these particular methodologies. This is clearer in section 2.2 but authors should specify with a higher degree of precision in the introduction why they believe these methods are suitable to treat the field of analysis. With a special reference to the DEMATEL application, the authors may like to check the following paper: Carpitella, S., Carpitella, F., Certa, A., Benítez, J., Izquierdo, J. Managing human factors to reduce organisational risk in industry. Mathematical and Computational Applications, 23 (4) (2018), 67, DOI: 10.3390/mca23040067. | Reasons related to the choice of the adopted methods have been incorporated in both in Introduction and more explicitly in Section 2.2. |
2. | Section 4.1 starts by describing the involved team of experts. It remains not clear to me if decision-makers have assigned all the same weights to get the aggregated pairwise comparison matrix (Table 10) for the AHP application. I recommend to specify how opinions were collected/aggregated and if experts have the same importance in the decision-making process or not, properly justifying the choice. | Proper explanations have been added in Sections 4.1 and 4.2.1. |
3. | In section 4.2 of results and discussion there are many tables in sequence reporting the detailed development of the various steps of the DEMATEL. I suggest to create an appendix and move there Tables 5,6,7 while keep in section 4.2 just one final table of results, integrating data of tables 8 and 9 into a single table. In this way the whole section would acquire a better look, broadly improving readability. Please, adjust also each caption of tables and check they are correctly referenced in the text. | Tables 5,6, and 7 are now moved to Appendix Section and renamed as Table A1, A2 and A3 respectively. Consequently, all other tables of Section 4.2 have been renumbered. |
4. | A part from the final ranking of factors, the output of the DEMATEL procedure is a causal diagram to graphically show interdependencies among elements (step 6, described in lines 232/244). Please add this chart to section 4.2 and comment results with relation to the specific position of factors in the quadrants of the chart. | Figure 2 shows the DEMATEL cause and effect diagram.
|
5. | The words of line 91 have to be separated. | Has been corrected. |
6. | The size of formulas 2/7 (from line 211 to 232) has to be reduced to be uniformed with formulas 9/11 (lines 261/264). Please do the same for formulas 12 and 13. | Has been corrected. |
7. | Table 4 has to be separated with spaces from the text above and below (lines 301/303). | Table 4 has been placed properly. |
8. | The general use of English has to be polished and carefully checked out throughout the whole manuscript. | English has been polished and carefully checked out throughout the entire manuscript. |
BEST
Reviewer 3 Report
The content of the paper is interesting but there are some major concerns.
The main question is: Why did you applied Best-worst method if you already have the results of AHP (and DEMATEL) method. Best-worst method is meant to be an easier substitution for the AHP method as described in [41]. Best-worst method demands less pairwise comparisons compared to AHP. But if you already have all pairwise comparisons for AHP method it is not clear why would you like to apply the best-worst method.
Why the results of pairwise comparisons of AHP and best-worst method differ? In Table 10 X1 to the others evaluations are: 1, 9, 2, 6, 2, 3, 5, 2, 4, while in Table 12 the evaluations are: 1, 4, 4, 3, 6, 2, 3, 7, 4. The compared criteria are the same, the scale 1-9 is the same, so the results should be the same.
DEMATEL:
How did you get the group evaluation from individual scores (in the application). If you used arithmetic mean of the scores, this should be mentioned in the description of method.
Page 7, lines 218 to 220 are redundant. If you want to have them in the paper than correct line 220: T=X(I-X)^-1 and describe what k means.
AHP:
page 8, line 260: a_ij=w_i/w_j
Equation (10) is redundant because A is usually not consistent. The method for deriving weights from the inconsistent matrix (with CR<0.1) should be described. You used additive normalization method in the application. The description of the method is missing.
How did you get the group evaluation from individual pair wise comparisons? This should be described in the description of the method.
RESULTS:
Page 11, line 371: The weights are also in Table 8. It is not clear what presents column 4 in Table 9. How did you get global weights in Table 9? If you multiplied local weights by weights of F1-F4, this should be described.
Page 13, line 387-389: This sentences are wrong. Your matrix in Table 10 is not consistent and you cannot use eq. (10).
Why didn’t you take into account factors F1-F4 in AHP method like you did in DEMATEL?
Minor remarks:
page 2, line 91: The spaces in the line are missing.
page 3, line 105: Number 2 is redundant.
page 6, line 202: A space before (A) is missing.
page 8, line 257: Comma instead of dot between i and j.
page 8, lines 259 and 262: A is consistent matrix (not consistency).
page 9, models (12) and (13) should be centralized or left aligned.
page 13 Replace commas by dots in decimal numbers.
Author Response
Here are the responses to the reviewers.
Sl. No. | Query | Response |
1. | The main question is: Why did you applied Best-worst method if you already have the results of AHP (and DEMATEL) method. Best-worst method is meant to be an easier substitution for the AHP method as described in [41]. Best-worst method demands less pairwise comparisons compared to AHP. But if you already have all pairwise comparisons for AHP method it is not clear why would you like to apply the best-worst method. | Reasons related to the choice of the adopted methods have been incorporated in both in Introduction and more explicitly in Section 2.2. |
2. | Why the results of pairwise comparisons of AHP and best-worst method differ? In Table 10 X1 to the others evaluations are: 1, 9, 2, 6, 2, 3, 5, 2, 4, while in Table 12 the evaluations are: 1, 4, 4, 3, 6, 2, 3, 7, 4. The compared criteria are the same, the scale 1-9 is the same, so the results should be the same. | Thank you for warning us about this issue. We went through again to the computational process and checked there were some miss calculations. It happened due to intense calculation volume. Therefore, with a very precise consideration, we recalculated BWM computation based on the new pairwise comparison (similar to AHP) and obtained the weights of decision factors. The revised Table is highlighted. It is interesting that the recent BWM computation enhances the consistency degree and is equal to 0,071 which is very acceptable and close to zero. We repeat that in the new BWM weighting process, we handled based on the AHP pairwise priority of x1 over other criteria and also the all the criteria over x2. All the relevant Tables are added in Appendix Section. |
3. | DEMATEL: How did you get the group evaluation from individual scores (in the application). If you used arithmetic mean of the scores, this should be mentioned in the description of method. Page 7, lines 218 to 220 are redundant. If you want to have them in the paper than correct line 220: T=X(I-X)^-1 and describe what k means. | Thank you for this comment; we have added this in Section 4.2.1 in the first paragraph.
All lines are corrected now. k is a value which is used in normalization process of DEMATEL. The expression of k is given by Eqn. (3).
|
4 | AHP: page 8, line 260: a_ij=w_i/w_j Equation (10) is redundant because A is usually not consistent. The method for deriving weights from the inconsistent matrix (with CR<0.1) should be described. You used additive normalization method in the application. The description of the method is missing. How did you get the group evaluation from individual pair wise comparisons? This should be described in the description of the method.
| - We have modified the Table for AHP pairwise comparison. The number of Tables are designed and highlighted to express the whole process. All relevant tables are added in Appendix Section.
The design of questionnaire was in this manner; initially we have requested experts to rate factors and sub-factors using DEMATEL individually, then we aggregated them using average of each expert value. Then, for the second step (AHP), it has been decided to use a different strategy to get the experts opinion. This time we asked them to all together agree on a unique questionnaire and do the pairwise comparisons. |
5 | RESULTS: Page 11, line 371: The weights are also in Table 8. It is not clear what presents column 4 in Table 9. How did you get global weights in Table 9? If you multiplied local weights by weights of F1-F4, this should be described. Page 13, line 387-389: This sentences are wrong. Your matrix in Table 10 is not consistent and you cannot use eq. (10). Why didn’t you take into account factors F1-F4 in AHP method like you did in DEMATEL? | The process of calculating global weights is added in Section 4.2.1.
All relevant tables are corrected and added. |
Minor remarks: page 2, line 91: The spaces in the line are missing. | Has been corrected. | |
page 3, line 105: Number 2 is redundant. | Has been corrected. | |
page 6, line 202: A space before (A) is missing. | Has been corrected. | |
page 8, line 257: Comma instead of dot between i and j. | Has been corrected. | |
page 8, lines 259 and 262: A is consistent matrix (not consistency). | Has been corrected. | |
page 9, models (12) and (13) should be centralized or left aligned. | Has been corrected. | |
page 13 Replace commas by dots in decimal numbers. | Has been corrected. |
best
Round 2
Reviewer 2 Report
The authors properly enhanced the manuscript and implemented all the required modifications. According to my opinion, the article is ready to be published.
Author Response
Many thanks for your consideration and giving us the chance of publishing in this journal.
Reviewer 3 Report
The main question in the first report was: Why did you applied Best-worst method if you already have the results of AHP (and DEMATEL) method. I am not satisfied with your answer. It is not explained why the weights of factors in Table 7 are not ok? Why you have to apply Best – worst method and why the weights of the factors in Table 10 are better or more correct?
Line 209: correct the equation
Lines 392-394: It is important to mention here that the condition (A − l max I)X = 0 must be met in order to achieve a consistent result while using AHP method.
This is true, but your matrix is not consistent!! You calculated the CR=0.09 in appendix. Delete this sentence.
Line 397: indicating the judgments made the experts are consistent acceptable
If CR=0, then matrix is consistent. If CR<0.1 then the matrix is acceptably consistent.
Some suggestions from the previous report has not been corrected nor commented:
Line 248, 251: If A is a consistency matrix correct: If A is a consistent matrix
Line 249: a_ij=w_i/w_j
Line 255: The method for deriving weights from the inconsistent matrix (with CR<0.1) should be described. You used additive normalization method in the application. The description of the method is missing.
Line 389-390: This is additive normlization method. Eq. 9 and 10 were not used. Correct!
Author Response
We, group of authors, thank you for your detailed comments and sensitivity to a qualified research article. Below we attach a file including the responses to the comments one by one. Hope this meets your doubts and requirements.

Round 3
Reviewer 3 Report
The paper was improved. I have only two minor remarks:
You stated: Statistical analysis of [45] has established that BWM accomplishes significantly better results than AHP with respect to the CR, minimum violation (preservative ability of ordinal preferences), total deviation (actual Euclidean distance between the ratios of any weights of any criteria pair and their corresponding pairwise comparisons) and conformity (intuitive evaluations by the DMs).
The reference [45] is not correct. It does not refer to AHP and BWM.
line 251: Equation a_ij=w_j/w_i is wrong. Swap i and j on the right hand of the equation. Correct equation: a_ij=w_i/w_j. This equation holds if and only if CR=0. This should be mentioned in the text.
Author Response
Thank you for giving us the comments for details. We appreciate the reviewer concern to our work and in attached file, the revised version is available.
